# A comparative study of causal perception in Guinea baboons (*Papio papio*) and human adults

Floor Meewis[1,2]*, Iris Barezzi[1], Joël Fagot[1,2], Nicolas Claidière[1,2], Isabelle Dautriche[1]

**1** Centre de Recherche en Psychologie et Neurosciences, UMR7077, Aix-Marseille University, CNRS, Marseille, France, **2** Station de Primatologie-Celphedia, CNRS UAR846, Rousset, France

* floormeewis@gmail.com

**Data Availability Statement:** Materials are available here: https://osf.io/vt5g8/?view_only= 5c8a8b7be8254d74ba1a4d9e4de6704e.

**Funding:** This work was supported by 1) the Agence Nationale de la Recherche (ANR-20-CE28-

## Abstract

In humans, simple 2D visual displays of launching events ("Michottean launches") can evoke the impression of causality. Direct launching events are regarded as causal, but similar events with a temporal and/or spatial gap between the movements of the two objects, as non-causal. This ability to distinguish between causal and non-causal events is perceptual in nature and develops early and preverbally in infancy. In the present study we investigated the evolutionary origins of this phenomenon and tested whether Guinea baboons (*Papio papio*) perceive causality in launching events. We used a novel paradigm which was designed to distinguish between the use of causality and the use of spatiotemporal properties. Our results indicate that Guinea baboons successfully discriminate between different Michottean events, but we did not find a learning advantage for a categorisation based on causality as was the case for human adults. Our results imply that, contrary to humans, baboons focused on the spatial and temporal gaps to achieve accurate categorisation, but not on causality *per se*. Understanding how animals perceive causality is important to figure out whether non-human animals comprehend events similarly to humans. Our study hints at a different manner of processing physical causality for Guinea baboons and human adults.

## Background

In humans, the impression of physical causality can be induced upon seeing simple 2D visual displays of moving geometric shapes [for review, see: 1]. When an object A moves towards a stationary object B, and if B starts moving as soon as A stops and touches B (a launching event, see Fig 1), people infer that A *caused* B to move. Introducing a temporal gap or spatial gap between A and B eliminates this impression of causality. This phenomenon has first been studied by Michotte in the 1940's [2] and it has inspired a wealth of research ever since.

Much debate has revolved around the mechanisms of causality inference and whether it involves deliberate and conscious cognitive mechanisms or can be explained as a perceptual process [e.g. 3, 4]. Michotte himself adhered to this latter interpretation (Michotte, 1946/1963)

0005, awarded to ID, https://anr.fr/Project-ANR-20-CE28-0005) and 2) the Institute for Language, Communication and the Brain (ILCB) through the Agence Nationale de la Recherche (ANR-16-CONV-0002, https://anr.fr/ProjetIA-16-CONV-0002). The funders had no role in study design, data collection and analysis, decision to publish, or preparation of the manuscript.

**Competing interests:** The authors have declared that no competing interests exist.

and recent studies demonstrate that–at least for simple launching events–causal perception is indeed a perceptual phenomenon [5–7].

Already at birth, babies show an attentional bias towards the continuous movements of direct launches compared to events with a temporal delay [8]. Newborns appear thus to be sensitive to the specific spatiotemporal features that differentiate between causal and non-causal events, but this does not necessarily mean they use causality to distinguish between them. This ability is suggested to develop later in life: habituation-dishabituation studies revealed that six-months-old infants, but not younger infants [but see: 9], group together different non-causal events, thereby categorically distinguishing causal from non-causal events [10, 11]. Consequently, there seems to be a developmental trajectory where younger infants employ a feature-based discrimination strategy, based on spatiotemporal cues (i.e., the presence or the absence of a spatial or a temporal gap), to distinguish Michottean events, which transforms into a category-based strategy, based on causality, from six months onwards. It is important to note that when actual toys are used as stimuli, this categorical distinction is only evidenced from ten months old [12]. This may be because the more realistic events distract infants from focussing on causality, suggesting that the use of simple and abstract figures is a strong method for studying causal perception [13].

Whilst the developmental origins of causal perception have received some attention, much less is known about its phylogenetic roots. Causal perception has been studied in only a limited number of non-human animal species using protocols that test the animals' sensitivity to violations of physical causality. For example, young Eurasian jays (*Garrulus glandarius*) and chimpanzees (*Pan troglodytes*) look longer at videos of "spatial gap events" where a food object is moved without contact [resp. 14, 15]. Similar experiments have been conducted on other species using 2D Michottean events. Newly-hatched chicks (*Gallus gallus*) have an innate predisposition to orient towards the launcher in causal direct launch, but such a preference does not exist for a non-causal temporal gap event [16]. Similarly, pet dogs (*Canis familiaris*) look longer and show more pupil dilation to a non-causal event with a spatial gap compared to a causal event (Völter & Huber, 2021). Similarly to looking preference designs in human infants, an important limitation of these studies is that they cannot disclose whether this discrimination is based solely on low-level features or (also) on the concept of causality. In a go/no-go task where pigeons (*Columba livia*) were trained to peck to discriminate between Michotte's causal and non-causal stimuli, the causal distinction was difficult to learn and instead the pigeons focussed on low-level features [17]. Therefore, it remains unclear whether non-human animals can experience an impression of causality with such stimuli. One study, however, revealed that

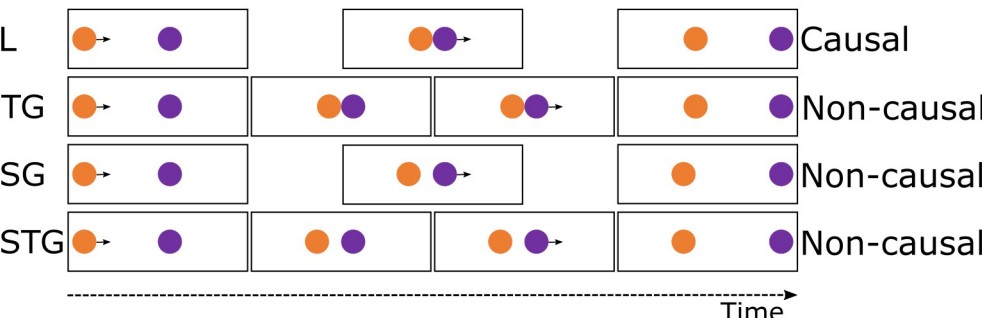

**Fig 1. Four different Michottean events.** Michottean events with the relevant event frames in temporal order from left to right. From top to bottom: Direct launching event (L), temporal gap event (TG), spatial gap event (SG), spatial +temporal gap event (STG). Only the direct launching event is regarded as causal, the other three as non-causal.

chimpanzees are susceptible to the 'causal capture effect' by showing that synchronous presentation of a causal event (bouncing) increases the experience of a second, ambiguous bounce/stream animation as causal [18]. However, this study did not use the same Michottean launching events as are usually presented in other studies on human adults and infants, preventing direct comparisons.

Extracting causality from ongoing events is at the core of how humans represent events [19, 20], but it is yet to be established whether non-human animals process the world around them in the same way [21]. A recent study showed that, similarly to humans, apes may have a bias towards animate agents in realistic videos, but causality was not specifically addressed [22]. Since the chimpanzee may be the only non-human species for whom causal perception for Michottean-like events is evidenced, it is relevant to investigate whether species with whom humans share a less recent common ancestor also perceive causality. For some of those species, it has been reported that they respond differently to causal and non-causal events (e.g. pet dogs), but there is no conclusive evidence whether this is because of the causal dimension. Therefore, to further investigate the evolutionary origins of causal perception with the classic Michottean events, we tested whether Guinea baboons (*Papio papio*) categorise these events based on causality. We subsequently tested human adults on the same categorisation task to compare the two species. Guinea baboons and humans share a common ancestor that lived about 30 Mya, so this comparison could, in concert with the previous findings in chimpanzees, shed light on the evolution of causal perception in a wider part of the primate lineage. Additionally, Guinea baboons are considered an excellent model species for studying human cognitive evolution [23–25] and are therefore a promising species to investigate causal perception. Because causal perception has been frequently documented in humans, shown to be early developing and perceptually processed, we hypothesised that both baboons and human adults would perceive causality and therefore use it to classify Michottean events.

In our study, we firstly trained the participants to distinguish a direct launch (Fig 1, top row) from a similar event which included a spatial and temporal gap (Fig 1, bottom row). Next, we introduced events with just a temporal (Fig 1, second row) or just a spatial gap (Fig 1, third row) and tested whether it was easier to categorise these events according to a division based on causality. We expected to find a faster learning rate for a categorisation based on causality compared to control, based on the assumption that the saliency of the causal events would set them apart from the non-causal events.

## Methods

Methods have been preregistered and materials are available here: https://osf.io/vt5g8/.

### Ethics statement

**Baboons.** The study on baboons was carried out in accordance with French and EU standards and received approval from the French Ministère de l'Education Nationale et de la Recherche (#APAFIS-2717-2015111708173794-V3). Procedures used in the present study were also consistent with the guidelines of the Association for the Study of Animal Behaviour. During this study, we collected behavioural data only. The study is non-invasive; no anaesthesia was administered, nor were the baboons sacrificed at the end of the study. Potential suffering is minimised as the baboons can partake in the experiments or not, as they please.

**Humans.** The study on human adults has received approval from the CEEI of INSERM (#20–733). We provided information about the task to the human participants and they consented to participate voluntarily in the study after having read the instructions by clicking on a button that said "I agree" before entering the online task.

## Participants

**Baboons.** Participants were 23 Guinea baboons (*Papio papio*, 16 females, mean age = 12.9 ± 1.2(SEM) years, age range = 4.5–23.1 years, see S1 Table in S1 File) who were housed at the Station de Primatologie in Rousset-sur-Arc (France) in two groups of eighteen and five individuals. The enclosure has a large outdoor area (700 m²) which is enriched with big stones as climbing structures, and a small indoor area which is accessible for shelter. The baboons are not water- or food-deprived: they always have access to water and are fed daily (monkey chow, fruits, vegetables). The health of the monkeys is monitored by the keepers and veterinarians.

For testing, we used the automatic learning device for monkeys (ALDM) developed by Fagot & Paleressompoulle [26]. With this system, the baboons had *ad libitum* access to testing booths with touch screens where they participated voluntarily in computerised tasks employing an operant conditioning method. This testing method, in which the baboons can come and go as they please, reduces stress levels compared to periods during which the ALDMs are not accessible [27]. The baboons have been tested with the ALDM for approximately ten years before the start of the current experiment, but they have never been exposed to Michottean events or any other video stimuli. Data collection took place in May and July 2022. For a pilot study held in January, we showed the small group of baboons (N = 5) Michottean events in a different task and with different stimuli. Additionally, a technical error prevented a smooth display of some of our stimuli of the first phase (see below) of the first stimulus set in May, after which we adjusted them and restarted our experiment.

**Humans.** On the 23rd and 24th of May 2023, we recruited 40 participants through the Prolific website, a platform where online experiments can be hosted. We set our criteria to accept only participants who reported to speak English fluently, so that they could understand the instructions. We paid participants £5 for their time.

We analysed the data from 39 participants (female = 21, male = 18; mean age = 27.5 ± 1.4 (SEM) years, age range = 20–59 years). Of the recruited people, we excluded one participant because they did not complete the experiment in one go, but paused for several hours in the middle. We had piloted our study in April/early May 2023 on twenty-one participants to gain insight into a possible effect size to estimate our needed sample size and to calculate the time needed for participants to finish the task to decide on appropriate participant payment.

## Procedure

We trained the monkeys and humans using a match-to-sample (MTS) paradigm to first discriminate and secondly categorise Michottean causal and non-causal events, see Fig 2. For each trial in the MTS task, a video was used as a sample, followed by two distinctive coloured shapes as comparison stimuli of which the participant had to touch one. The sample video was displayed in the centre of the screen, and the comparison stimuli were displayed for the monkeys in its bottom part and for the humans in the middle, with their right-left location counterbalanced across trials. The comparison stimuli were used as arbitrary depictions of the categories and which response stimulus was assigned to which event category was counterbalanced across participants. Training took place by giving feedback to the participants. For the monkeys, we rewarded correct responses with some wheat grains, whereas incorrect responses were punished with a 3s green time-out screen. Touching the screen during the video interrupted the trial and also led to a 3s green time-out screen. The humans received a 1s green screen with the word 'Yes!' to correct responses and a 3s red screen with the word 'No!' to incorrect responses. We presented the trials in random order in blocks and determined the

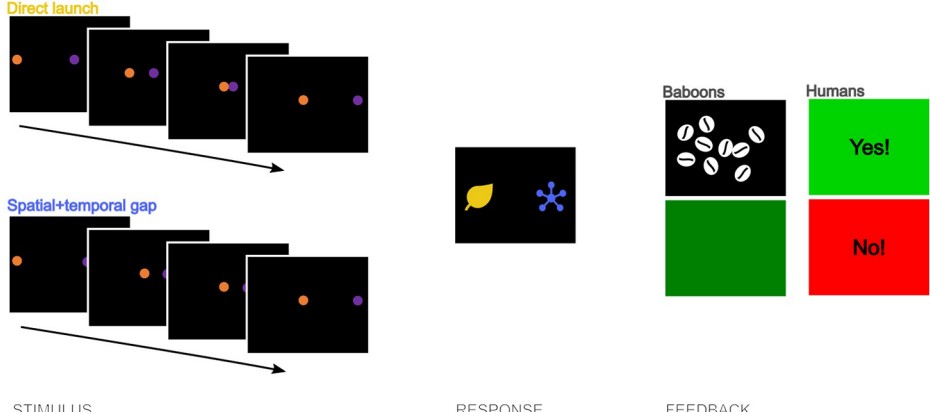

**Fig 2. Match-to-sample paradigm.** The participant (baboon or human) views one of the Michottean events play out, which is followed by a screen with two response stimuli. Upon pressing one of the buttons, the participants receive feedback; for the baboons, correct results in a food reward and incorrect in a 3s green time out screen, for the humans, correct leads to a green screen with 'Yes!' and incorrect to a 3s red screen with 'No!'.

classification accuracy by calculating the percentage of correct responses for each block. The baboons had blocks of sixty trials and the humans blocks of eight trials in the training phases.

The humans received minimal instructions before the start of the experiment. We explained that they would see videos which they had to classify by pressing one of two buttons. We also told them they would receive feedback in some of the trials (as a green or red screen) and in some other trials no feedback in the form of a purple screen. Lastly, we told them that the task's duration would increase with wrong answers, to give them an incentive to try to answer correctly. Causality was not mentioned. After the experiment had finished, the human participants were asked to answer to a question on how they had categorised the videos. We did this to test whether people would explicitly mention causality (of the 39 participants, 13 mentioned that some videos were causal [8 times in the causal and 5 times in the control condition]). Also, we asked about their age, gender, nationality and first language.

## Stimuli

The stimuli were based on Michotte's causal and non-causal events [1,2; see Fig 1]. We used four event types: the direct launch (L), the temporal gap event (TG), the spatial gap event (SG) and the spatial and temporal gap event (STG). The temporal gap (present in TG and STG) lasted always for one second and the spatial gap (present in SG and STG) was always a quarter of the area between the start of object A and the end of object B. Note that the STG event always had both of these gaps and the L event had none. Multiple variants were created for each event category in which we changed the speed of the objects and the length of the trajectories of objects A and B, making the defining features of each event type the presence or absence of the gap(s), see S2 and S3 Tables in S1 File. All videos were between 800 and 2600ms. We used two distinct stimulus sets. The first stimulus set consisted of videos depicting an orange circle as object A and a purple circle as object B with a movement direction from left to right. The second stimulus set used a blue square as object A and a yellow square as object B with a movement direction from right to left. Within each stimulus set, the colours of object A and B are used consistently regardless of event type (L, SG, TG and STG). Stimuli were presented on a black background. The videos were made using the animation features of Microsoft® PowerPoint® and subsequently exported in a mp4 format with a resolution of

944x 720 pixels and a frame rate of 30 frames/s. Videos and response stimuli were presented to the baboons using E-prime (V2) and to the humans using Labvanced [28]. All videos are available on OSF (in the folder named Stimuli).

## Experimental phases and conditions

The experiment consisted of four phases. An overview of the experiment is presented in Fig 3. The participants started with phase 1 and continued in the experiment according to their performance. The baboons participated two times in the experiment to do both conditions, each with a different stimulus set. The humans were randomly assigned to a condition and a stimulus set.

**Phase 1: L vs. STG.**   First, we trained the participants to discriminate one L from one STG video, see videos 1 and 2 in S2A Table in S1 File. These two events were matched for their total duration (i.e., 1600 ms). Participants were presented with blocks of randomly ordered trials, such that, within a block, there were half of L trials and half of STG trials. To continue to the next phase of the experiment, the baboons had to reach 80% correct classification for both

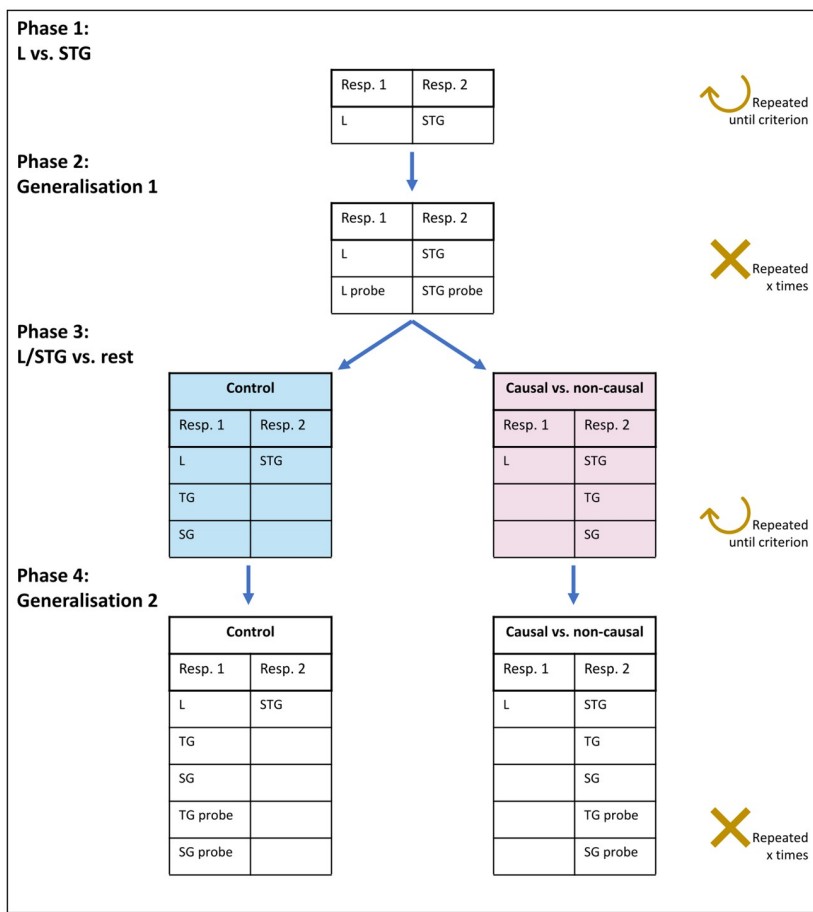

**Fig 3. Overview of the experimental phases.** The Michottean events that are grouped together under the same response stimulus (resp. 1 or resp. 2) are shown. The participants were trained to discriminate L from STG in Phase 1 and tested on their transfer ability to new probe events in Phase 2. In Phase 3, TG and SG events were added and the participants were trained according to a control (in blue) or causal vs. non-causal (in pink) condition. Transfer to new probe SG and TG events was tested in Phase 4.

events within the block (60 trials) and the humans had to reach 100% correct classification for all the trials in the block (8 trials).

After reaching these pre-set criteria, two extra events were added, one L and one STG, see videos 3 and 4 in S2B Table in S1 File. With these two events, we controlled for distinction strategies that employed the following 6 'irrelevant' features: (1) speed, (2) durations, (3) distances of movements of the objects, (4) initial and (5) end positions of the objects, and (6) total duration of the events (see S3 Table in S1 File). That is, if a participant tried to distinguish the events based on one of the above-mentioned features, they would not succeed. We therefore trained the participants to classify using a strategy which could involve causality or paying attention to the presence or absence of the gaps. Each block consisted of 25% of previously learnt L events, 25% of previously learnt STG events, 25% new L events and 25% new STG events. When the baboons reached the criteria of 70% correct on all four events, they continued to the next phase.

**Phase 2: Generalisation 1.** To verify whether the participants had not reached high accuracy scores by just memorising the four training videos and how they associate with the response stimuli (which they had seen many times throughout training), we tested whether they could transfer their learnt categorisation to new L and STG events.Here, we tested the performance of the participants on four probe events that they had not seen before: two L and two STG events, see videos 5, 6, 7 and 8 in S2C Table in S1 File. These probes were different in terms of durations, speed, position of the point of impact, initial position of object B and end position of object A, compared to the training events.

For the baboons, we used blocks of 60 trials, including 8 probe trials (4 L; 4 STG) and 52 previously learnt Phase 1 trials (26 L and 26 STG), and repeated these blocks sixteen times. The probes were randomly rewarded in 80% of cases regardless of the response given to avoid any learning for these trials. For the humans, we presented three times a block of six trials composed of the four probe events, and the two previously learnt events from the first training phase. We did not give the humans feedback on these probe trials, but instead showed a purple screen for two seconds after they made their choice, regardless of the response given, which was explained in the instructions as no feedback.

Before going to phase 3, we also tested the participants' reaction to Michottean events of which the direction had been reversed. The same method as for phase 2 generalisation 1 was employed using these reversed events as probes. This test was designed for a separate study and will not be discussed further in the current paper.

**Phase 3: L/STG vs. rest.** This crucial phase tested whether a categorisation based on causality would lead to a learning advantage compared to a categorisation which did not involve causality. We trained the participants to classify events with just a spatial gap (SG) and just a temporal gap (TG), into their previously learnt categories either forming a causal vs. non-causal division or a control division. Half of the trials of each block were comprised of the four previously learnt L and STG events and half of the new SG and TG events (each 25%), see videos 9 and 10 in S2D Table in S1 File.

In the causal vs. non-causal condition, the participants had to learn to press the same response stimulus for the SG and TG events as for the STG event (see Fig 3, Phase 3, right), effectively dividing causal from non-causal events. In the control condition, participants had to press the same response stimulus for SG and TG as for the L event (see Fig 3, Phase 3, left), leading to a mix of causal and non-causal events. Critically, both the SG and the TG events are equally different from the STG event as from the L event in terms of their perceptual features [presence or absence of a gap; see 10]. This means that learning to group together TG and SG with STG in the causal vs. non-causal condition implies a difference of the size of the spatial gap (always ¼ of the screen) and the size of the temporal gap (1s) and exactly these

differences are also present for grouping them with L. Since we do not know whether the spatial gap and temporal gap are of equal salience, we decided to divide the classification such that the participants always had to learn a rule that implicated both the temporal and the spatial domain.

The monkeys were alternately assigned to one of the conditions (causal vs. non-causal or control) upon reaching phase 3. A few months later, we ran the second part of the experiment with a new stimulus set; here, the monkeys were admitted to the condition they had not participated in. The humans were randomly assigned to one condition and one stimulus set.

For both conditions, we measured the learning speed as the number of blocks that were needed to reach a pre-determined learning criterion. For the baboons we set this at 70% correct per event type (L, STG, SG and TG) in a block within a maximum of 35 blocks, and for the humans we set it at 100% correct for the whole block.

**Phase 4: Generalisation 2.** Lastly, we tested whether the categorisation learnt in Phase 3 would transfer to new, unseen SG and TG stimuli. We added two new SG and two new TG events which had different speeds and durations compared to the training videos, see videos 11, 12, 13 and 14 in S2E Table in S1 File. For the baboons, we introduced eight probe videos per block, which were randomly rewarded in 80% of cases, and they were intermingled with 52 trials of previously learnt events (17 L, 17 STG, 9 SG and 9 TG). We repeated the block sixteen times. For the humans, we showed three times a block of six trials composed of the four probe events and one L and one STG event from Phase 1 with the two-second purple screen as uninformative feedback.

## Statistical analyses

All statistical analyses were conducted in R [29].

**Phase 1 (learning L vs. STG).** we did not conduct any statistical analyses, but we calculated how many blocks were needed to reach our pre-set learning criteria for both the baboons and humans. Additionally for the baboons, we tracked which individuals passed the criteria.

**Phase 2 and Phase 4 (generalisation phases).** For each generalisation phase, we implemented a generalised linear mixed model with binomial error structure using the lme4-package [30] to model the effect of trial type (baseline or probe) and the effect of event type (L, STG, SG, TG) on trial accuracy (0/1). We also checked whether the stimulus set (first or second) would affect the accuracy. Adding stimulus set was not preregistered, but we wanted to verify whether the different stimuli would not affect the performance. For the second generalisation phase (Phase 4), we also implemented post-hoc an interaction between the condition (causal or control) and event type, because the distribution of responses for the event types depended on the condition (see Fig 2, Phase 3). We allowed the intercepts to vary per participant. We tested the main effects using likelihood ratio tests.

For Phase 2, the first generalisation phase, we used the following model: trial accuracy ~ trial type + event type + stimulus set + (1|participant).

For Phase 4, the second generalisation phase, we used for the baboons the following model: trial accuracy ~ trial type + event type * condition + (1|participant). We have thus removed the stimulus set, because it did not improve the model fit. Post-hoc, we tested whether it is easier to tell apart L from the other events in the causal vs. non-causal condition compared to the STG in the control condition by using contrasts of the estimated marginal means using the 'emmeans' package [31]. For the humans, the model with interaction between condition and event type did not converge, so we continued with a model without this interaction. Additionally, we removed trial type, since the probes and baseline had non-overlapping event types.

We tested for Phase 2 and Phase 4 whether the percentage of correct responses per event type was above 50% (chance level) by using the intercept values of the model summary and changing the baseline categories accordingly.

**Phase 3 (control vs. causal/non-causal conditions).** We compared the learning speed for the causal vs. non-causal compared to the control condition, i.e., how many blocks of trials are needed to reach the learning criteria in each condition. For the baboons, we used a within-subjects design and we tested whether there was a difference in learning speed between the two conditions using a non-parametric sign-test. This method deviates from what was anticipated in the preregistration and was implemented because the intended model with random intercepts per participant did not converge as too few individuals participated in both conditions (N = 6). Additionally, the upper limit of 35 blocks to reach the learning criteria prevented us from accurately determining the shape of the distribution and apply standard parametric tests. To compare learning speeds of the control and causal division in the human data, we implemented a between-subjects design and we used a Mann Whitney U-test because the data did not show equal variances and normal distributions. Finally, we checked the responses of the human participants to see whether there was mention of causality in their reported strategies.

## Results

### Phase 1: L vs. STG

Twelve baboons learnt to discriminate one L from one STG event for at least one of the stimulus sets, which took on average 53.1 ± 12.3(SEM) blocks of sixty trials. Subsequently, these baboons continued to the next step in the experiment where we added two more events (one L and one STG). From these four training videos, eleven baboons learnt to discriminate L from STG in on average 36.0 ± 10.2(SEM) blocks, see S1 Table in S1 File for the individual performances. All human participants learnt to discriminate L from STG in two videos after on average 4.5 ± 1.1(SEM) blocks of eight trials and continued in the experiment and again learnt the division for four videos after 2.4 ± 0.5(SEM) blocks.

### Phase 2: Generalisation 1

The eleven baboons performed above chance on categorising the newly introduced probe events, for both L and STG, as well as for the previously learnt training events of L and STG (all < .001, all mean accuracy above 62.0±3.4(SEM)%). Additionally, there was an effect of stimulus set on accuracy ($\chi^2$ = 9.44, p = .002); generalisation was more accurate for the first (72.0±1.8(SEM)%) than the second stimulus set (68.9±1.8(SEM)%). Also, our model showed an effect of trial type ($\chi^2$ = 60.66, p < .001) showing that baboons performed more accurately on the training events (73.0±1.4(SEM)%) than on the probe events (67.3±2.2(SEM)%). Event type did not influence accuracy (L vs. STG, $\chi^2$ = 1.35, p = .245).

For the humans, the data of generalisation 1 showed that the participants performed very accurately on all event types and trial types, such that our final sample included 47 incorrect to 655 correct responses. To check whether this excess of 1's would pose a problem, we tested for possible "1-inflation" using the DHARMa package [32] which indicated no issues (p = .612) and led us to continue with our original binomial model. Our model showed no effect of trial type ($\chi^2$ = 0.63, p = .429) and no effect of event type ($\chi^2$ = 0.20, p = .656). They performed significantly above chance on all shown events (all p < .001, all mean accuracy above 91.45±2.9 (SEM)%).

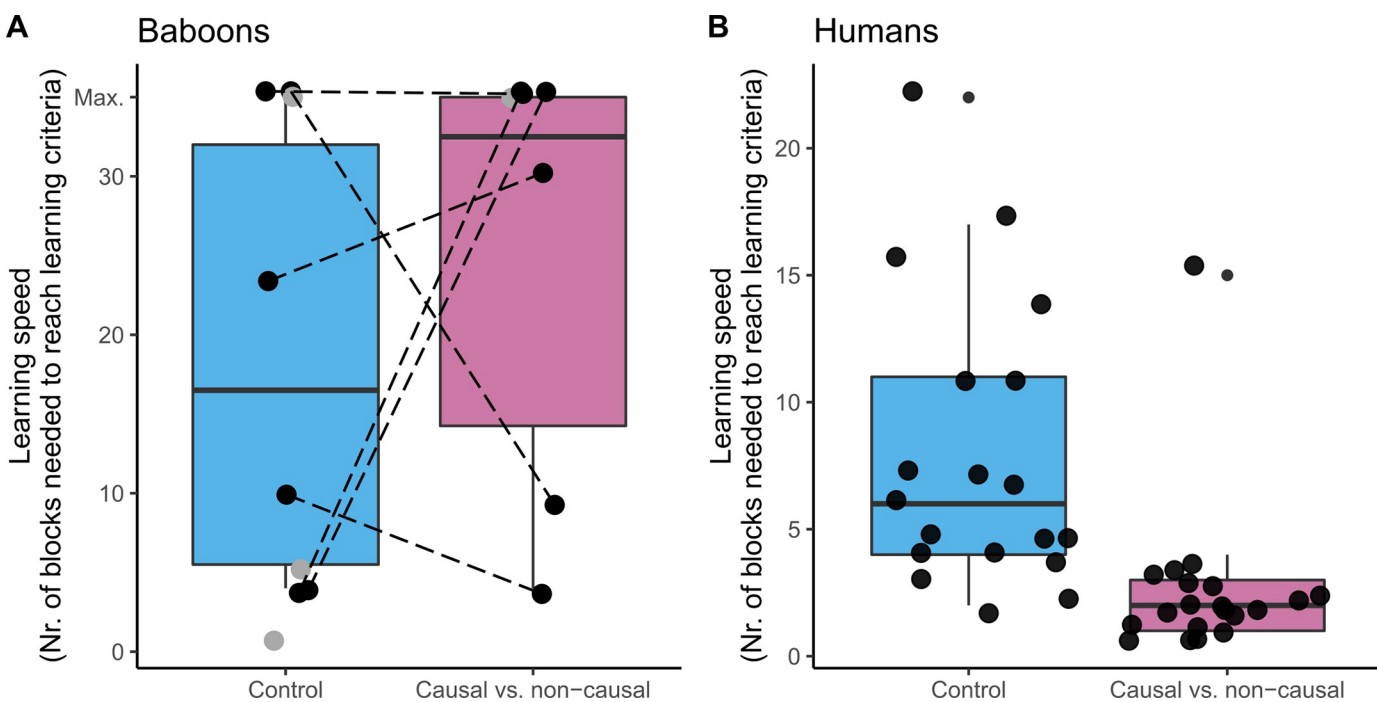

**Fig 4. Baboons' and humans' learning speed to learn the control and causal vs. non-causal condition.** Learning speed is expressed as number of blocks of trials needed to reach our learning criteria. (A) The baboons were equally fast to learn the control as the causal vs. non-causal condition. Dotted lines connect individuals who have participated in both conditions shown in black, grey dots are individuals who participated in one condition and do not make part of the statistical test. (B) The humans were faster to learn the causal condition than the control condition.

### Phase 3: L/STG vs. rest

Of the 11 baboons who participated in this phase in at least one of the conditions, 5 participated in both of the conditions and managed to reach our learning criterion in at least one condition. There was no significant difference between the speed at which baboons learnt the causal condition (mean = 24.7 ± 5.8(SEM) blocks) and the speed at which they learnt the control condition (mean = 18.5 ± 5.9(SEM) blocks; N = 5, successes = 2, p = 1), see Fig 4.

All human participants learnt the condition they were assigned to. Humans were significantly faster to learn the causal condition (2.7±0.7(SEM)) compared to the control condition (8.0±1.3(SEM); N = 39, W = 38.5, p < .001).

### Phase 4: Generalisation 2

For the baboons, our model showed a significant effect of trial type ($\chi^2 = 25.00$, p < .001) such that baseline trials resulted in higher accuracy than probe trials (80.0±2.6(SEM)% for baseline and 77.0±3.1(SEM)% for probe). Also, we found an interaction effect between condition and event type ($\chi^2 = 843.56$, p < .001). Further exploration showed that the participants performed significantly above chance on the newly introduced probe events (all < .001, all mean accuracy above 66.1±6.7(SEM)%), but, on the previously learnt baseline trials the performance on the launching event (L) in the causal vs. non-causal condition did not differ significantly from the 50% chance level (p = .05, mean accuracy = 54.8 ±10.0(SEM)%), see Fig 5. Accuracy on all other baseline event categories for both conditions remained significantly above chance level (all p < .001, all mean accuracy above 68.0±5.5(SEM)%). To check whether the L event stands out from the others, our post-hoc analysis showed that for the event category that had to be

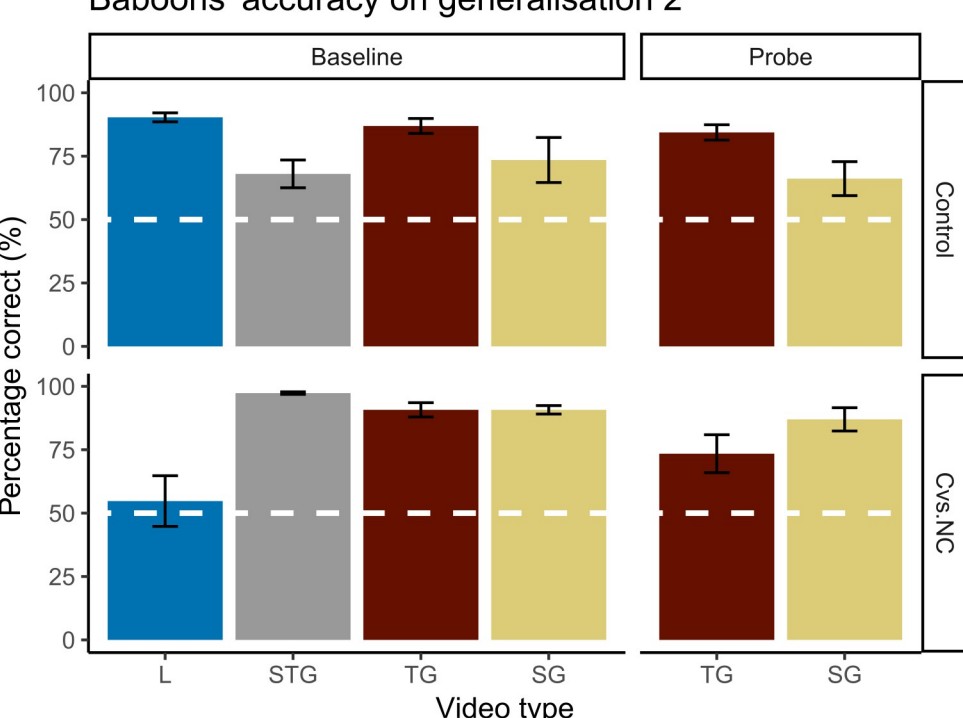

**Fig 5. Baboons' performance on generalisation 2.** The y-axis shows the percentage of correct classification, with the chance level of 50% correct marked in white. The different Michottean event types are displayed on the x-axis. On top the control condition and at the bottom the causal vs. non-causal condition.

classified separately from the other three (i.e., L in the causal vs. non-causal condition and STG in the control condition), baboons were worse at classifying launching events in the causal vs. non-causal condition than classifying the STG event in the control condition (54.8 ±10.0(SEM)% vs. 68.0±5.5(SEM)%; estimate = 0.498, SE = 0.227, p < .001).

On the human data, there was a significant effect of condition ($\chi^2$ = 14.12, p < .001), indicating that the participants in the causal vs. non-causal condition were more accurate (93.3 ±1.7(SEM)%) than participants in the control condition (84.4±2.9(SEM)%). Additionally, there was an effect of event type ($\chi^2$ = 20.95, p < .001), reflecting that the performance on SG events (81.2±4.4(SEM)%) was lower than the three other event types (L = 91.5±2.7(SEM)%, STG = 89.7±3.7(SEM)% and TG 93.6±2.2(SEM)%). Importantly, all event types had an accuracy that was significantly above chance level (all p < .001, all mean accuracy above 64.04±7.13 (SEM)%), demonstrating that the humans successfully categorised previously unseen SG and TG events.

## Discussion

We introduced a new paradigm to assess whether individuals favour causality or spatiotemporal properties when presented with Michottean events. In this categorisation task, Guinea baboons were as successful to learn to categorize Michottean events following a causal division (causal vs. non-causal) as they were to categorize these events according to a control division that was equal in terms of spatiotemporal differences. By contrast, humans were faster in the causal categorisation condition. This showed that, even though both species

demonstrated the ability to successfully distinguish and categorise Michottean events and generalise to novel events, only the humans made use of causality. Guinea baboons relied on spatiotemporal properties only to solve the categorisation task. Even though we cannot rule out that Guinea baboons have causal representations of Michottean events, we show that where humans apply causality to sort Michottean events, Guinea baboons rather use a feature-based discrimination.

One could argue that the 2D stimuli may be too distant from any natural experience of causality baboons routinely encounter, and therefore they may not distil a causal relation from the depicted events. There are two reasons to doubt that the use of artificial stimuli is the reason why the baboons did not categorise according to causality in our task. First, in infant studies, where it is common practice to employ simplified stimuli [cf. 13], causal perception has been evidenced in very young infants using similar 2D shapes [10, 11, 33]. Similarly to non-human animals, 6-month-old infants' experience with animated shapes is limited. As such, these stimuli are as unnatural and unfamiliar for baboons as for infants, suggesting that it is unlikely that experience with the stimuli may explain a difference in performance between these two populations. We believe that the baboons who participated in our study may actually have *more* experience engaging with on-screen stimuli than infants, because of their involvement in other studies using artificial stimuli in the same computerised set-up [e.g. 34]. Secondly, even when demonstrations with real-life 3D objects are employed, monkeys sometimes have difficulty to use contact-mechanical information over spatiotemporal features, as was shown for rhesus macaques (*Macaca mulatta*) who failed to search in the right location for food which had been stopped from rolling by a barrier [35]. Instead, the rhesus performed better on tasks that centred around finding the food using spatiotemporal information, a point that we come back to below. Because of these reasons, we think that the use of 2D stimuli is appropriate for baboons and not responsible for their absence of use of causality.

We note, however, that our categorisation task was challenging for the baboons to learn as only 7 of 11 baboons were able to reach our learning criteria in phase 3, where they had to learn to categorise spatial gap events and temporal gap events, suggesting difficulties in the task at this point. Note that the other baboons of the group who did not pass phase 1 (learning to distinguish a direct launch (L) from a spatial and temporal gap event (STG)), may have done so for various reasons, such as lack of interest, not necessarily because they did not or could not understand the task. The difficulties during phase 3, can presumably be explained by the discrepancy of correct responses between the two comparison stimuli. In phase 3, baboons had to learn to answer one response stimulus in a quarter of the trials, and the other response stimulus in the remaining three quarters of the trials. This discrepancy has driven some baboons towards a response strategy where one response stimulus is touched all the time regardless of the Michottean event stimulus, leading to a 75% reward pay-off with minimal effort. This is also visible in the subsequent generalisation phase: the baboons' performance to set apart the causal event in the causal vs. non-causal condition worsened to chance level and was even lower than their ability to set apart a spatial and temporal gap event in the control condition. This suggests that the causal nature of the launching event is not an outstanding feature for the baboons. In contrast, the humans perform well despite this difficulty, indicating that causality is something that they pick up on with ease. Overall, it is clear that the causal/non-causal division has not been salient enough for the baboon participants to override this bias towards one response stimulus.

Besides causality, what other strategies for successful classification were at the baboons' disposal? Certainly, baboons could not have used superficial features of our stimuli, such as the speed of the objects or the duration of the events, since our choice of stimuli controlled for those. To learn the classifications of the two conditions equally fast, the only possible way is to

rely on the presence or absence of the temporal and spatial gap. Such a categorisation strategy where focussing on a combination of features instead of a concept can account for successful classification, is in line with the feature model for discrimination [e.g. in pigeons, 36]. In our task, this points to using the addition of spatial and temporal properties instead of using causality (based on these features). The deployment of a feature-based strategy finds support in previous work which shows that, in other tasks, Guinea baboons tend to prioritise local features over global aspects of presented stimuli [e.g. 37–39]. Other species, such as rhesus macaques, similarly demonstrate a preference to rely on spatiotemporal features over contact information when parsing (dynamic) mechanical relations [35]. In sum, our results point to Guinea baboons employing a feature-based strategy, based on the presence or absence of a gap, to categorise Michottean events.

Other non-human animal species have also demonstrated discriminatory abilities between different Michottean events in previous studies (newly-hatched chicks: 16,dogs: 39), but, contrary to our design, these studies did not include ways to specifically test whether the animals employed causality or only spatiotemporal cues for their discrimination. The evidenced discrimination by the chicks as shown by a preference for the launcher in causal launches and no preference in temporal gap events, appeared to be driven primarily by the self-propelled motion of the first object [16], leaving open the question whether causality or the presence or absence of the temporal gap had played a role as well. Also the dogs might preferentially look towards the non-causal event [40], because of differences in the spatiotemporal properties of the stimuli, instead of causality *per se*. These studies thus show discriminatory abilities similar to what we have demonstrated in the first two phases of our experiment for eleven baboons, but we additionally implemented the specific test for sensitivity to causality. Similarly, Young *et al.* [17] trained pigeons with a go/no-go paradigm to peck on one of four events displayed on a screen (L, SG, TG or STG), with the expectation that if pigeons discriminate the events based causality rather than spatiotemporal features, the individual trained to peck for the causal launching event should find the task easiest. However, even after extensive training, just one of four pigeons was able to generalise the event they learnt to peck on, to novel stimuli, whereas the other pigeons were not able to pick up on the relevant spatiotemporal properties nor on causality. Consequently, for all these non-human animals, evidence favouring the use of causality over low-level properties is still lacking.

Currently, chimpanzees appear to be the only species that is sensitive to causality in Michottean-like events [18]. How can we interpret the different results obtained between chimpanzees and, here, Guinea baboons? One possibility is that this difference stems from the use of different paradigms (causal capture effect for the chimpanzees vs. categorisation task for the baboons). Another possibility is that causal perception has evolved sometime after the separation between the Cercopithecoidea and the Hominoidea ~30Mya. This could explain why causal perception can be observed in humans and chimpanzees, but not on other species. Nonetheless, our study demonstrated a sensitivity to the specific spatiotemporal properties necessary for causality detection in Guinea baboons, suggesting that this may be an evolutionary trait that is shared between Guinea baboons and humans.

Our findings for Guinea baboons clearly differ from those observed for the human adults, who were much faster to learn the task and who showed a clear advantage to learn the causal vs. non-causal division. Indeed our findings support the idea that, for human adults, the distinction between causal and non-causal events is a central and prominent feature of perception. It has been reported numerous times that humans possess a capacity to quickly detect causality from visual displays [1] and we show this again with a novel paradigm. The causal vs. non-causal distinction could be seen as a mental default and has even been argued to underlie the occurrence of causative structures cross-linguistically [41].

Contrary to human adults' ability to use causality when presented with Michottean launches, we believe the baboons' findings to be more in line with how human infants younger than six months of age react to Michottean launches. Cohen and Amsel [10] showed with a habituation-dishabituation paradigm that it was only six-, but not four- and five-month olds, who did not dishabituate upon seeing a new non-causal event when they had been habituated to another. This suggest that infants younger than six months of age do not make use of causality yet, even though they are able to detect (spatiotemporal) differences between the stimuli from birth [8]. It is clear, however, that the looking time methodology employed for human babies is different from the categorisation used in our study. It may be the case that an implicit looking time experiment is more sensitive to test a perceptual phenomenon than a categorisation task. Future studies could exploit eye-tracking or habituation paradigms to try and corroborate our hypothesis that Guinea baboons react similarly to infants younger than 6 months of age when presented with Michottean events.

Even though our findings indicate that Guinea baboons may not perceive causality (or at least do not use it for categorization) after being probed with Michottean launches, this does not necessarily imply that they lack the ability to detect causal structures altogether. Certainly, various animal species have been shown causal reasoning abilities, at least to a certain extent [42–44], and it is possible that these abilities are more likely to be detected by monkeys in actions/events that involve animate agents [45, 46]. However, our studies point to a critical distinction between humans and Guinea baboons in their ability to transform visual sequences of motions with certain spatiotemporal parameters into a causal impression.To conclude, causality detection is needed to understand events [20] and in particular the interrelatedness of the entities involved [47]. Extracting these event elements and how they relate to one another from ongoing experiences is crucial in understanding the world around us [48]. Investigating whether non-human animals process events similarly to humans provides insights into the evolutionary roots of event cognition [21, 22]. Understanding how animals perceive causality is an important starting point to figure out whether non-human animals comprehend events similarly to humans. For causality perception in Michottean events, our study hints at a different manner of processing for Guinea baboons and human adults.

## Supporting information

**S1 File.** This file contains all supporting information including 'S1 Table. Baboon details.', 'S2 Tables. Stimuli details.' and 'S3 Table. Categorisation strategies.'.
(DOCX)

## Acknowledgments

We thank Julie Gullstrand, Sebastien Barniaud and the staff at the Station de Primatologie for their help with the experiments with the baboons.

## Author Contributions

**Conceptualization:** Floor Meewis, Iris Barezzi, Joël Fagot, Nicolas Claidière, Isabelle Dautriche.

**Data curation:** Floor Meewis, Iris Barezzi.

**Formal analysis:** Floor Meewis, Iris Barezzi.

**Funding acquisition:** Isabelle Dautriche.

**Investigation:** Floor Meewis, Iris Barezzi, Nicolas Claidière, Isabelle Dautriche.

**Methodology:** Floor Meewis, Iris Barezzi, Nicolas Claidière, Isabelle Dautriche.

**Project administration:** Floor Meewis, Isabelle Dautriche.

**Resources:** Joël Fagot, Nicolas Claidière, Isabelle Dautriche.

**Supervision:** Nicolas Claidière, Isabelle Dautriche.

**Visualization:** Floor Meewis.

**Writing – original draft:** Floor Meewis, Isabelle Dautriche.

**Writing – review & editing:** Floor Meewis, Joël Fagot, Nicolas Claidière, Isabelle Dautriche.

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
