## [Decision Letter · Decision Letter 0]

30 Jul 2024

PONE-D-24-17807A comparative study of causal perception in Guinea baboons (*Papio papio*) and human adultsPLOS ONE

Dear Dr. Meewis,

Thank you for submitting your manuscript to PLOS ONE. After careful consideration, we feel that it has merit but does not fully meet PLOS ONE’s publication criteria as it currently stands. Therefore, we invite you to submit a revised version of the manuscript that addresses the points raised during the review process.

I received three reviews from experts in the field.  Reviewers 1 and 2 have a very positive view of the manuscript.  Reviewer 3 generally has a positive view of the manuscript, but also lists several comments that should be addressed in a revision.  If you feel you can successfully respond to the comments made by the reviewers, then I invite you to submit a revision, along with a response to each comment raised by the reviewers.

We look forward to receiving your revised manuscript.

Kind regards,

Darrell A. Worthy, Ph.D

Academic Editor

PLOS ONE

Journal Requirements:

"This work was supported by the Agence Nationale de la Recherche (ANR-20-CE28-0005) and the Institute for Language, Communication and the Brain (ILCB) through the Agence Nationale de la Recherche (ANR-16-CONV-0002). We are thankful to Julie Gullstrand, Sebastien Barniaud and the staff at the Station de Primatologie for their help with the experiments with the baboons. "

"This work was supported by

1) the Agence Nationale de la Recherche (ANR-20-CE28-0005, awarded to ID, https://anr.fr/Project-ANR-20-CE28-0005) 

and 2) the Institute for Language, Communication and the Brain (ILCB) through the Agence Nationale de la Recherche (ANR-16-CONV-0002, https://anr.fr/ProjetIA-16-CONV-0002). 

Reviewers' comments:

Reviewer's Responses to Questions

**Comments to the Author**

1. Is the manuscript technically sound, and do the data support the conclusions?

Reviewer #1: Yes

Reviewer #2: Yes

Reviewer #3: Yes

2. Has the statistical analysis been performed appropriately and rigorously? 

Reviewer #1: Yes

Reviewer #2: Yes

Reviewer #3: Yes

3. Have the authors made all data underlying the findings in their manuscript fully available?

Reviewer #1: Yes

Reviewer #2: Yes

Reviewer #3: Yes

4. Is the manuscript presented in an intelligible fashion and written in standard English?

Reviewer #1: Yes

Reviewer #2: Yes

Reviewer #3: Yes

5. Review Comments to the Author

Reviewer #1: This manuscript explores the details of causal perception in Guinea baboons and humans. The context of the study is clearly framed, and the important distinction between causal perception and production (e.g. some tool-use tasks) is highlighted. It is clearly written and the experimental design is well-controlled. The conclusion is supported by the results and various alternative explanations are considered in the Discussion. Particularly important is the distinction between spatio-temporal categorisation and true causality, the latter only achieved by humans and not baboons. This paradigm can be used for many different species (after training on touchscreens), which will hopefully provide additional insights into the evolution of causal perception.

Reviewer #2: This study gives very compelling evidence that baboons do not systematically categorize causal launching interactions as different from non-causal interactions, and instead categorize events based only on low-level spatiotemporal features. I have no complaints about the method or analyses. The data are also very clearly presented, and while some of the results are a bit perplexing (notably that baboons did slightly better categorizing STG vs. everything else), these are considered in the discussion. The discussion appropriately addresses all of the most salient concerns about the experiment. I would personally have liked to see more discussion of the distinction between "making a distinction based on low-level features" and "understanding launching as a causal interaction", and possible ways in which you could have the latter without the former, but I am also satisfied with the report as it stands since it is an issue that must be left for later work.

Reviewer #3: This paper outlines a novel approach to understanding the evolutionary roots of causal attribution. The authors use a match-to-sample touchscreen task to test baboons and humans, finding that humans easily distinguish causal from non-causal events, whereas baboons demonstrate similar performance between a causal/non-causal distinction and a control condition (where causal and non-causal are mixed).

The introduction is concisely written and the goals of the study are clear. I also find the study itself well thought out by requiring participants to form either causal/non-causal classifications, or classifications based off a mix of spatial and temporal information. I did find the methods, however, hard to follow without rummaging around in the supplementary material. Readers should be able to understand how the study was done without having to refer to the ESM in great detail. I have included some points below which I found were not clearly explained. I also include some suggestions for the discussion which address why there might these species differences.

Methods

Were the colours in the stimulus set counterbalanced between roles? It would be important to rule out that responses were not just being made from low level features, such as agents being orange.

Line 122-132. It’s not clear how the choice of symbols relates to the event viewed, and how the participants knew which symbol to choose in which trials or how this demonstrated their ability to classify events.

How many times did participants view each video?

Discussion

The authors discuss the use of artificial stimuli as appropriate for studying causal detection in baboons. My concern is not so much with the stimuli themselves (the authors make a good justification for their use) but with the type of causality that is presented. Previous work in macaques indicates that they may be sensitive only to familiar actions. How familiar are baboons with launching events? To what extent do the authors expect that these results would generalise to other actions (for example, a chasing scenario).

e.g.

Familiar actions:

Rochat et al. 2008.The evolution of social cognition: Goal familiarity shapes monkeys’ action understanding. Current Biology, 18, 227-232. https://doi.org/10.1016/j.cub.2007.12.021

Wood et al. 2008. Rhesus monkeys’ understanding of actions and goals. Soc. Neurosci. 3, 60–68. https://doi.org/10.1080/17470910701563442

Geometric chasing events:

Galazka & Nystrom, 2016. Infants’ preference for individual agents within chasing interactions. J. Exp. Child Psychol. 147, 53–70. https://doi.org/10.1016/j.jecp.2016.02.010

I find it an unlikely argument that baboons and other NHPs beyond apes can’t detect causality, given that these are species living in complex social groups with the need to track third party interactions. It is more likely that there are qualitative differences in these species’ abilities to detect causality, possibly being driven by divergent demands on social or spatial cognition. This point also relates to the above comment on action sensitivity/relevance.

OSF

The description for some of the videos in the ESM includes: ‘Also, we will test if the baboons are sensitive to event roles, known as agent and patient, which are present in causal, but not non-causal, events.’ However, this is not really touched on in the paper, except in the last paragraph. Event roles in general are not explained. If the authors wish to address event cognition in this ms, I suggest they already introduce these concepts in the introduction. The manuscript does, however, stand on its own without these additional justifications.

Minor comments

Line 38-39. The cited study (Mascalzoni et al) used looking time to indicate that infants differentiate between launching event types. Whilst I appreciate that the authors themselves use the term ‘preference’ as an interpretation of these results, there have been a number of studies since then that question the use of this terminology to interpret what can only strictly be considered an attentional bias. It would be preferable, in light of this, that the authors use the term attentional bias instead of attentional preference in this manuscript.

e.g.

Tafreshi, Thompson and Racine 2014. An analysis of the conceptual foundations of the infant preferential looking paradigm. Human Development,57,222-240. DOI: 10.1159/000363487

Wilson, Bethell & Nawroth 2023. The use of gaze to study cognition: limitations, solutions and applications to animal welfare. Frontiers in Psychology, 14:1147278.

doi: 10.3389/fpsyg.2023.1147278

Winters, Dubuc & Higham 2015. Perspectives: The looking time experimental paradigm in studies of animal visual perception and cognition. Ethology, 121, 625-640. doi: 10.1111/eth.12378

Line 44. I don’t understand this sentence ‘the presence of the absence of a spatial or a temporal gap’.

Line 65-66. There is some preliminary evidence that addresses this question from a recently published study:

Brocard et al. 2024. iScience, 27,109996. https://doi.org/10.1016/j.isci.2024.109996

Line. 102. Should read ‘they always have’.

Line 108. Reduces stress levels compared to what?

Line 133. 23rd and 24th

Line 115. Should be £5

Line 169. Please provide sub-folder name so it’s easy to find the videos.

Line 177. It could be helpful to include a (supplementary) table that specifies number of trials and which videos are designed to control for each of these features.

Line 378. Should read ‘similar to’.

Line 400. ‘Contrary to this ability to human adults’ ability’ awkward wording.

References. Ref 18 – correct species italics.

6. PLOS authors have the option to publish the peer review history of their article (what does this mean?). If published, this will include your full peer review and any attached files.

Reviewer #1: No

Reviewer #2: **Yes: **Jonathan F. Kominsky

Reviewer #3: No

---

## [Author Response · Author response to Decision Letter 0]

6 Sep 2024

Reviewer #1 

This manuscript explores the details of causal perception in Guinea baboons and humans. The context of the study is clearly framed, and the important distinction between causal perception and production (e.g. some tool-use tasks) is highlighted. It is clearly written and the experimental design is well-controlled. The conclusion is supported by the results and various alternative explanations are considered in the Discussion. Particularly important is the distinction between spatio-temporal categorisation and true causality, the latter only achieved by humans and not baboons. This paradigm can be used for many different species (after training on touchscreens), which will hopefully provide additional insights into the evolution of causal perception.

>We are grateful for the positive evaluation by this reviewer and the potential they see in our paradigm. 

Reviewer #2 

This study gives very compelling evidence that baboons do not systematically categorize causal launching interactions as different from non-causal interactions, and instead categorize events based only on low-level spatiotemporal features. I have no complaints about the method or analyses. The data are also very clearly presented, and while some of the results are a bit perplexing (notably that baboons did slightly better categorizing STG vs. everything else), these are considered in the discussion. The discussion appropriately addresses all of the most salient concerns about the experiment. I would personally have liked to see more discussion of the distinction between "making a distinction based on low-level features" and "understanding launching as a causal interaction", and possible ways in which you could have the latter without the former, but I am also satisfied with the report as it stands since it is an issue that must be left for later work.

>We thank Dr. Kominsky for his favourable feedback. 

>Considering the distinction between the use of causality and the use of spatiotemporal features that he points out, we agree that this would be relevant to dive further into. We view causal perception of a launching event as something that goes beyond the sensitivity to spatiotemporal features, yet we do not think it is possible without basing this on low-level features because the two are inherently intertwined. We have now additionally stressed this in l. 341, l. 379 and l. 387.

Reviewer #3

This paper outlines a novel approach to understanding the evolutionary roots of causal attribution. The authors use a match-to-sample touchscreen task to test baboons and humans, finding that humans easily distinguish causal from non-causal events, whereas baboons demonstrate similar performance between a causal/non-causal distinction and a control condition (where causal and non-causal are mixed).

The introduction is concisely written and the goals of the study are clear. I also find the study itself well thought out by requiring participants to form either causal/non-causal classifications, or classifications based off a mix of spatial and temporal information. I did find the methods, however, hard to follow without rummaging around in the supplementary material. Readers should be able to understand how the study was done without having to refer to the ESM in great detail. I have included some points below which I found were not clearly explained. I also include some suggestions for the discussion which address why there might these species differences.

>We thank the reviewer for their thoughtful and constructive feedback. We will address all suggestions in greater detail below.

Methods

Were the colours in the stimulus set counterbalanced between roles? It would be important to rule out that responses were not just being made from low level features, such as agents being orange.

>We agree that it is important to keep alternative explanations based on features other than causality or the spatial and temporal gaps in mind, but we think the colours of the objects cannot have influenced the categorisation. Critically, we systematically kept the colours the same for all of the four events (L, SG, TG and STG) in the stimulus sets throughout the experiment and for all participants. In set 1, object A was orange and object B purple with a movement from left to right. In set 2, object A was blue and object B was yellow with a movement from right to left. Since the colours of the objects were the same across the events (and independently of their causal reality), we assume that colour did not help them in any way. We have now stressed the colour use in l. 166 “Within each stimulus set, the colours of object A and B are used consistently regardless of event type (L, SG, TG and STG).”

We did, however, counterbalance the response buttons in two ways: 1) We counterbalanced which event categories were assigned to which response button. For example, take the ‘L vs. rest’-condition with stimulus set 1; half of the participants had to classify L to the leaf response button and the SG, TG and STG to the molecule response button, whereas the other half had to place L with the molecule and SG, TG and STG with the leaf. We did the same for stimulus set 2 and the ‘STG vs. rest’-condition. 2) We counterbalanced the left-right location of the response stimuli across trials (sometimes the leaf would appear on the left and the molecule on the right, on other trials this was swapped). This is noted in the procedure section of the methods, l.129-132.

Line 122-132. It’s not clear how the choice of symbols relates to the event viewed, and how the participants knew which symbol to choose in which trials or how this demonstrated their ability to classify events.

How many times did participants view each video?

>We are happy to clarify this part of the methods. Indeed, as the reviewer appears to suspect, the response buttons do not have any a priori relation to the events. The participants are therefore trained by operant conditioning to form a predefined, but arbitrary association between the event and the response stimulus. In short, the buttons are just a means to establish classification and their shape or colour are not meant to bear clues to which is correct. We have clarified this in the text as follows: “The comparison stimuli were used as arbitrary depictions of the categories and which response stimulus was assigned to which event category was counterbalanced across participants.”, l. 131.

The ability to classify the events is demonstrated by the percentage of correct responses. Since the participants start out without knowing which response shape is assigned to which event, their random responses will lead to an average accuracy of 50%. If they are able to learn the classification, they should thus score at least significantly above 50% correct; we set a criterion at 80 and 70% for the baboons and at 100% for the humans. We have adjusted our writing as follows: “We presented the trials in random order in blocks and determined the classification accuracy by calculating the percentage of correct responses for each block.”, l. 137.

Concerning the question of how often the participants saw the videos: in each trial the participants saw the event just once. However, throughout the experiment they saw the events many times, depending on how fast they learned the association. We have noted the number of blocks (of 60 trials for baboons and 8 trials for humans) needed to reach our learning criteria in the result section; this corresponds to the number of Michottean events seen throughout that phase of the experiment. Because the participants see the training events so often, we checked if they had not just memorised them to succeed in categorisation task in the generalisation phases (which was not the case). We have now underscored this: “To verify whether the participants had not reached high accuracy scores by just memorising the four training videos and how they associate with the response stimuli (which they had seen many times throughout training), we tested whether they could transfer their learnt categorisation to new L and STG events.”, l. 192.

Discussion

The authors discuss the use of artificial stimuli as appropriate for studying causal detection in baboons. My concern is not so much with the stimuli themselves (the authors make a good justification for their use) but with the type of causality that is presented. Previous work in macaques indicates that they may be sensitive only to familiar actions. How familiar are baboons with launching events? To what extent do the authors expect that these results would generalise to other actions (for example, a chasing scenario).

e.g.

Familiar actions:

Rochat et al. 2008.The evolution of social cognition: Goal familiarity shapes monkeys’ action understanding. Current Biology, 18, 227-232. https://doi.org/10.1016/j.cub.2007.12.021

Wood et al. 2008. Rhesus monkeys’ understanding of actions and goals. Soc. Neurosci. 3, 60–68. https://doi.org/10.1080/17470910701563442

Geometric chasing events:

Galazka & Nystrom, 2016. Infants’ preference for individual agents within chasing interactions. J. Exp. Child Psychol. 147, 53–70. https://doi.org/10.1016/j.jecp.2016.02.010

>We thank the reviewer for presenting these suggestions and relevant papers. Considering the potential role of (un)familiarity of the stimuli, we estimate that the baboons were presumably unfamiliar with launching events, at least with events where one object launches another one, and they had never seen schematic launching events on the screens before. Again, such unfamiliarity seems to be similar to the studies with human infants, so we do not expect that to have an effect specific to baboons, we have added this now in l. 350. 

We agree on the other hand that causal detection may depend on the type of causality that we presented the participants with. Here we showed the baboons a physical / mechanical causal relation (no animacy cues present), which can perhaps explain the differences with the studies that are mentioned here, which are all in the social realm with animate agents performing actions. Previous studies have shown difficulties in monkeys to focus on contact-mechanical information and global features of stimuli, as we laid out in l. 353-357 and l. 377-384. We can speculate that perhaps additional cues of animacy may facilitate the perception of a relation. In line with this, we have conducted another study that was partly based on the paper that is mentioned on chasing events with 2D stimuli (currently submitted). We show that baboons have an attentional bias towards the agent in chasing interactions (see our abstract: https://escholarship.org/uc/item/75n3n3g0), suggesting that they perceive this as a causal relation. We have now added the proposal about the animacy cues in the discussion, l. 424-429. 

I find it an unlikely argument that baboons and other NHPs beyond apes can’t detect causality, given that these are species living in complex social groups with the need to track third party interactions. It is more likely that there are qualitative differences in these species’ abilities to detect causality, possibly being driven by divergent demands on social or spatial cognition. This point also relates to the above comment on action sensitivity/relevance.

>We agree completely with the reviewer. In the current paper we do not claim that baboons cannot detect causality at all; rather that causality is either not perceived in Michottean launches or at least not used for the categorisation. The focus of this study is on causal perception of Michottean events, and we thus remain cautious in extending to other causal relations or causal reasoning. We have now made this point more explicit in the discussion, l. 424-429. Additionally, we have emphasized the point that our findings are specific to Michottean launches in our final conclusion, l. 434.

OSF

The description for some of the videos in the ESM includes: ‘Also, we will test if the baboons are sensitive to event roles, known as agent and patient, which are present in causal, but not non-causal, events.’ However, this is not really touched on in the paper, except in the last paragraph. Event roles in general are not explained. If the authors wish to address event cognition in this ms, I suggest they already introduce these concepts in the introduction. The manuscript does, however, stand on its own without these additional justifications.

>The reviewer is right that, additionally to the work presented in this manuscript, we tested whether the baboons would attribute agent and patient roles to the launcher and launchee by using the role reversal method developed by Leslie & Keeble (1987). We think this question deserves a more specific report with an additional - and slightly different - body of background literature. To be able to dive into this question more profoundly we decided to combine this data with an extra experiment we have conducted with human infants and not to include it in the current manuscript. We feel it would be more powerful to have all of this data together. We will remove this phrase from the description on OSF to avoid confusion. 

Minor comments

Line 38-39. The cited study (Mascalzoni et al) used looking time to indicate that infants differentiate between launching event types. Whilst I appreciate that the authors themselves use the term ‘preference’ as an interpretation of these results, there have been a number of studies since then that question the use of this terminology to interpret what can only strictly be considered an attentional bias. It would be preferable, in light of this, that the authors use the term attentional bias instead of attentional preference in this manuscript.

e.g.

Tafreshi, Thompson and Racine 2014. An analysis of the conceptual foundations of the infant preferential looking paradigm. Human Development,57,222-240. DOI: 10.1159/000363487

Wilson, Bethell & Nawroth 2023. The use of gaze to study cognition: limitations, solutions and applications to animal welfare. Frontiers in Psychology, 14:1147278. doi: 10.3389/fpsyg.2023.1147278

Winters, Dubuc & Higham 2015. Perspectives: The looking time experimental paradigm in studies of animal visual perception and cognition. Ethology, 121, 625-640. doi: 10.1111/eth.12378

>Thank you for pointing us to these relevant papers. We agree that the more parsimonious phrasing of attentional bias is better suited and we have changed the writing accordingly. 

Line 44. I don’t understand this sentence ‘the presence of the absence of a spatial or a temporal gap’.

>There was a typo there, thank you for noting that. We have changed it to “the presence or the absence of a spatial or a temporal gap”.

Line 65-66. There is some preliminary evidence that addresses this question from a recently published study:

Brocard et al. 2024. iScience, 27,109996. https://doi.org/10.1016/j.isci.2024.109996

>We agree this study is indeed very relevant. We have added this citation in the new version of the manuscript, l. 66. 

>Thank you for having noted the following typos and small errors, we have adjusted them accordingly. 

Line. 102. Should read ‘they always have’.

Line 108. Reduces stress levels compared to what?

>Compared to periods during which the ALDMs are not accessible, we have adjusted this in the text (l.111). 

Line 133. 23rd and 24th

Line 115. Should be £5

Line 169. Please provide sub-folder name so it’s easy to find the videos.

Line 177. It could be helpful to include a (supplementary) table that specifies number of trials and which videos are designed to control for each of these features.

>We had already included such tables in the supplementary materials, but we have updated the referencing to make them easier to find. 

Line 378. Should read ‘similar to’.

Line 400. ‘Contrary to this ability to human adults’ ability’ awkward wording.

References. Ref 18 – correct species italics.

>We are thankful to the reviewer for appreciating our work and for raising important issues which we hope to have successfully addressed in the revised manuscript.

---

## [Editor Report · Decision Letter 1]

17 Sep 2024

A comparative study of causal perception in Guinea baboons (*Papio papio*) and human adults

PONE-D-24-17807R1

Dear Dr. Meewis,

We’re pleased to inform you that your manuscript has been judged scientifically suitable for publication and will be formally accepted for publication once it meets all outstanding technical requirements.

I think you have adequately addressed the concerns raised by the reviewers in your revised manuscript.

Kind regards,

Darrell A. Worthy, Ph.D

Academic Editor

PLOS ONE
---

## [Editor Report · Acceptance letter]

24 Sep 2024

PONE-D-24-17807R1 

PLOS ONE

Dear Dr. Meewis, 

I'm pleased to inform you that your manuscript has been deemed suitable for publication in PLOS ONE. Congratulations! Your manuscript is now being handed over to our production team.

Kind regards, 

on behalf of

Dr. Darrell A. Worthy 

Academic Editor

PLOS ONE